# The Application of Adipose Tissue-Derived Mesenchymal Stem Cells (ADMSCs) and a Twin-Herb Formula to the Rodent Wound Healing Model: Use Alone or Together?

**DOI:** 10.3390/ijms24021372

**Published:** 2023-01-10

**Authors:** Hui Ma, Wing-Sum Siu, Chi-Man Koon, Xiao-Xiao Wu, Xiang Li, Wen Cheng, Wai-Ting Shum, Clara Bik-San Lau, Chun-Kwok Wong, Ping-Chung Leung

**Affiliations:** 1Institute of Chinese Medicine, The Chinese University of Hong Kong, Shatin, Hong Kong SAR, China; 2State Key Laboratory of Research on Bioactivities and Clinical Applications of Medicinal Plants, The Chinese University of Hong Kong, Shatin, New Territories, Hong Kong SAR, China; 3Department of Chemical Pathology, The Chinese University of Hong Kong, Hong Kong SAR, China

**Keywords:** MSCs, NF3, wound healing, Chinese medicine, stem cells

## Abstract

Our previous study reported that mesenchymal stem cells (MSCs) accelerated the wound healing process through anti-inflammatory, anti-apoptotic, and pro-angiogenetic effects in a rodent skin excision model. NF3 is a twin-herb formula, which presents similar effects in promoting wound healing. Research focusing on the interaction of MSCs and Chinese medicine is limited. In this study, we applied MSCs and the twin-herb formula to the wound healing model and investigated their interactions. Wound healing was improved in all treatment groups (MSCs only, NF3 only, and MSCs + NF3). The combined therapy further enhanced the effect: more GFP-labelled ADMSCs, collagen I and collagen III expression, Sox9 positive cells, and CD31 positive cells, along with less ED-1 positive cells, were detected; the expressions of proinflammatory cytokine IL-6 and TNF-α were downregulated; and the expression of anti-inflammatory cytokine IL-10 was upregulated. In vitro, NF3 promoted the cell viability and proliferation ability of MSCs, and a higher concentration of protein was detected in the NF3-treated supernatant. A proteomic analysis showed there were 15 and 22 proteins in the supernatants of normal ADMSCs and NF3-treated ADMSCs, respectively. After PCR validation, the expressions of 11 related genes were upregulated. The results of a western blot suggested that the TGFβ/Smad and Wnt pathways were related to the therapeutic effects of the combined treatment. Our study suggests for the first time that NF3 enhanced the therapeutic effect of MSCs in the wound healing model and the TGFβ/Smad and Wnt pathways were related to the procedure.

## 1. Introduction

Wound healing is a complex and dynamic process, involving a cascade of cellular events. In a normal condition, wound repair begins with cell migration (hemostasis), inflammation, proliferation, and extracellular matrix deposition (remodeling) [1]. The process can be compromised by multiple factors, including infection, ischemia, and intensive skin loss [2]. Acute wounds, such as surgical incisions, thermos wounds, and lacerations, cause a disturbance in the integrity of normal skin, which heals uneventfully. Chronic wounds are classified as instances when the wound fails to heal and remains open for more than one month [2]. In the United States, the Medicare cost projections for wounds ranged from USD 28.1 to 96.8 billion over the past four years, among which surgical wounds and diabetic ulcers formed the largest proportion [3]. Many factors can lead to prolonged wound healing, including age (65 years or older), underlying metabolic disease (such as hypertension and diabetes), malignancy, and obesity [3]. Since the COVID-19 pandemic, recent studies have demonstrated a higher risk of infection for patients with chronic wounds [4].

The current standard wound care procedure includes cleaning, dressing. and debridement [5]. For chronic wounds, the most commonly used methods are split-thickness autografts and cultured epithelial autografts (CEA). However, the side effects, including pain, donor-site infection, and long preparation time, limit their uses [5,6]. Wound dressings are used to preserve rehydration in the wound, protect against infection, and maintain the integrity of the wound base [7]. They are commonly made of chitosan, hyaluronic acid, collagen, and silicon. Other biomaterials that consist of alginates, heparin, cellulose, and gelatin are being investigated in both pre-clinical and clinical studies [7]. Other approaches to wound healing include the use of plant extracts, herbal medicines, and modern medicines. However, these are limited by the low efficacy and unfavorable side effects [8]. Therefore, a method to maximize the healing effect with minimal side effects is needed. 

Mesenchymal stem cells (MSCs), first isolated in the 1970s, are a group of progenitor cells of mesodermal source. They present a high self-renewing capacity and various linage differentiation abilities [9]. They are widely used in the study of wound care management, bone and cartilage diseases, cardiovascular and neurological diseases, and immune-/inflammation-mediated diseases [10]. However, there are many factors that limit their uses, such as poor engraftment, low survival rate, and donor-dependent variation of the cells [11]. Hence, strategies to enhance their functions are under investigating. These strategies can be categorized into preconditioning, genetic manipulation, use of supportive materials, and co-administration with other therapeutic methods [11]. The paracrine activity is the main underlying mechanism of MSC-based cell therapy. To enhance the function of MSCs, methods that can adjust their paracrine activity are key. 

NF3 is an innovative formula from our institute. It consists of two herbs: Astragali Radix and Rehmanniae Radix, with a ratio of 2:1. The formula has been shown to enhance the healing of diabetic ulcers by promoting angiogenesis and the suppression of inflammation through in vitro, in vivo, and clinical studies [12,13]. Research on the co-administration of Chinese herbs and stem cells is limited. We hypothesized that the therapeutic effects could be enhanced when using NF3 and MSCs together through the alternation of their paracrine activities.

In this study, we applied NF3 and MSCs to a rodent wound healing model and investigated the therapeutic effects and the interactions between them. 

## 2. Results

### 2.1. MSCs and NF3 Accelerated Wound Healing Speed

On Day 5 after the combined treatment (MSCs + NF3), wound size was significantly reduced (*p* < 0.05). From Day 7 to Day 14, wound healing was promoted in all groups (*p* < 0.05). The combined treatment further enhanced the healing effect (*p* < 0.05). No significance was observed between the MSCs group and the NF3 group. Wounds healed completely after 28 days (Figure 1).

### 2.2. MSCs and NF3 Increased Collagen Formation, Local Stem Cell Activation and Angiogenesis, Reduced Inflammation

As presented in Figure 2A, both NF3 and MSCs increased the expression of collagen I and collagen III, while the combined treatment further promoted the collagen formation. Since no injected MSCs were found after 14 days, only tissue collected on Day 14 was stained with Sox 9 (stem cell marker). The data suggested that both MSCs and the combined treatment increased the number of Sox 9-positive cells (*p* < 0.05). No significance was observed in the NF3 group (Figure 2B). All treatments presented the pro-angiogenic effect on Day 7, and the highest number of CD 31 positive cells was observed in the combined treatment group (*p* < 0.05). On Day 14, the number of CD 31 positive cells only increased in the MSCs and the combined groups (*p* < 0.05), while no difference was found in the NF3 group (Figure 2C). The number of inflammatory cells was reduced remarkably after different treatments (*p* < 0.05). No difference was observed between the MSCs and NF3 groups. On Day 7, fewer ED-1 positive cells were found in the combined group than the MSCs group (*p* < 0.05), while no difference was indicated between the NF3 and combined groups. The fewest ED-1 positive cells were found in combined group on Day 14 (*p* < 0.05) (Figure 2D). 

### 2.3. The Survival Rate and Survival Time of the Injected MSCs Was Increased by NF3 Administration

Skin was collected daily after injection to exam the expression of GFP. The GFP expression peaked on Day 4 and diminished gradually from Day 5. In the combined treatment group, there was remarkably more GFP expression from Day 3 to Day 9 (*p* < 0.05). No significant changes were observed after 10 days. No GFP expression was detected on Day 14 (Figure 3). 

### 2.4. MSCs and NF3 Regulated the Expression of Inflammation-Related Genes and Extracellular Matrix (ECM) Related Genes

On Day 7, the expression of IL-1β was downregulated by the MSCs administration. Both MSCs and the combined treatment reduced the expression of IL-6 and TNF-α. The expression of the anti-inflammatory gene IL-10 was significantly upregulated in all groups (*p* < 0.05). The expressions of MMP-1 and MMP-2 were downregulated in the NF3 group, but upregulated in the MSCs and combined treatment groups (*p* < 0.05). The expression of TIMP1 was reduced in all groups (*p* < 0.05). The ratio of MMP1 to TIMP1 was calculated. It increased in all groups, but compared with MSCs, both NF3 and the combined treatment reduced the ratio (*p* < 0.05) (Figure 4A). 

Figure 4B presents the data from Day 14. Similar to the data on Day 7, only MSCs reduced the expression of IL-1β (*p* < 0.05). The level of TNF-α was reduced significantly in all groups (*p* < 0.05). No difference was observed in the expressions of IL-6 or IL-10 from any of the groups. The expression of MMP-1 was upregulated in the MSCs group only. The expressions of MMP-2 and TIMP1 were reduced significantly in all groups (*p* < 0.05). The ratio of MMP1 to TIMP1 was increased, but it was lower in the MSCs and the combined treatment groups (*p* < 0.05) (Figure 4B). 

### 2.5. The Cell Viability, Proliferation, and Mobility of MSCs Were Enhanced by NF3

MSCs were identified via flow cytometry and the induction of tri-lineage differentiation. As previously described, they expressed CD29 and CD90, but were lacking in CD45. They also presented the differentiation potential in adipocytes, chondrocytes, and osteocytes (Appendix A) [9,14].

Different doses of NF3 (0.025–2 mg/mL) were added to the MSCs cultivation. The data suggested that high-dose NF3 increased the cell viability (1–2 mg/mL) and proliferation (0.25–2 mg/mL) of MSCs (*p* < 0.05). No significant differences were observed in other doses (Figure 5A,B). In the scratch test, only the medium dose of NF3 (0.1–0.25 mg/mL) promoted wound closure on the plate (*p* < 0.05). No significant differences were observed in other doses (Figure 5C).

### 2.6. NF3 Modulated the Proteomic Profile of MSCs’ Supernatant

After being treated with different doses of NF3, the total protein concentration in the serum-free supernatant was measured. The results showed that a higher concentration of protein was detected in the supernatant treated with high dose of NF3 (0.2–2 mg/mL) (*p* < 0.05) (Figure 6A). A total of 79 proteins were examined in the supernatants after NF3 administration. There were seven and twenty-two proteins detected in the untreated supernatant and NF3-treated supernatants, respectively. There were 15 unique proteins only found in the NF3-treated supernatant (Figure 6B). The function related to the wound healing of all proteins are listed in Table 1. After calculating the mean pixel density, the results suggested that the densities of the most common proteins were significantly higher in the NF3-treated supernatant (Figure 6C). RT-PCR was used for validation. Samples from different doses (0.25–2 mg/mL) were tested. The expressions of 11 corresponding genes were upregulated remarkably, and most of them presented in a dose-dependent manner (Figure 6D). 

### 2.7. The Wnt/β-Catenin and TGFβ-1/Smad 2/3 Pathways Are the Potential Targets

Two pathways related to the stemness of stem cells and the wound healing process were investigated. The data suggested that NF3 (2 mg/mL) increased the expressions of Wnt2, TGFβ-1, and Smad2/3 significantly (*p* < 0.05) (Figure 7).

### 2.8. NF3 Promoted the Proliferation and Mobility of Dermal Fibroblasts (FR)

Different doses of NF3 were also added to dermal fibroblasts (FR). No cytotoxic effect was shown in any of the doses (Figure 8A). A medium dose of NF3 (0.1–0.5 mg/mL) promoted the proliferation of NF3, while a high dose (1–2 mg/mL) reduced the ability (*p* < 0.05) (Figure 8B). In a scratch test, the data indicated that a high dose of NF3 (0.25–2 mg/mL) enhanced the mobility of FR significantly (*p* < 0.05) (Figure 8C). 

### 2.9. NF3 Reduced the Levels of Nitric Oxide (NO), IL-6, MCP-1, and TNF-α Released from RAW 264.7

Different doses of NF3 were added to RAW264.7 (macrophage) to evaluate the effect of anti-inflammation. A high dose of NF3 (0.25–2 mg/mL) produced a significant cytotoxic effect, so only a low dose was used in the following tests (*p* < 0.05) (Figure 9A). NF3 (0.2 mg/mL) reduced the level of nitric oxide (NO) in the supernatant after LPS activation (*p* < 0.05) (Figure 9B). The concentrations of IL-6, MCP-1, and TNF-α are presented in Figure 9C–E, respectively. The results showed that the levels of all pro-inflammatory cytokines were decreased significantly in a dose-dependent manner (*p* < 0.05) (Figure 9C–E).

## 3. Discussion

### 3.1. Effect of MSCs and NF3 on the Wound Healing Model

Wound healing, which consists of four precisely and highly programmed phases, is a normal biological process. To achieve the process successfully, the four phases need to occur in a proper sequence and time frame [15]. To provide more detail, the following events happen in the wound healing process: (i) hemostasis; (ii) inflammation; (iii) mesenchymal cell differentiation, proliferation, and migration to the wound site; (iv) angiogenesis; (v) re-epithelialization; (vi) synthesis, cross-linking, and the alignment of collagen to provide strength to the healing tissue [15]. Many factors can affect the regulated process, such as aging, infection, obesity, diabetes, and ischemia. Both conventional and biological methods are used to enhance wound healing. In contrast to conventional wound management, biological methods can target multiple pathological events, and not only protect against infection but also promote angiogenesis and re-epithelialization [9]. 

MSCs have become a therapeutic option in recent decades. As shown in our study, both MSCs and NF3, when used as individual therapies, promoted wound healing in the first two weeks. Several studies have indicated that MSCs could function in all of the four phases, throughout the whole healing process. Firstly, one study reported that there were high expressions of phosphatidylserine and tissue factor on the surfaces of MSCs, which could trigger coagulation [16]. In wound healing, this may contribute to promoting rapid hemostasis. Secondly, MSCs are well-known in regard to the modulation of inflammation. The inflammatory phase is critical in determining a normal or impaired wound healing. MSCs secrete several growth factors and cytokines to regulate neutrophils, macrophages, and lymphocytes. Their anti-inflammatory effect has been demonstrated in different animal models [9,14,17]. In wound healing, the anti-inflammatory effects are achieved through the polarization of macrophages from the pro-inflammatory M1 phase to the anti-inflammatory M2 phase [18]. In our study, a pan-marker of macrophages, ED-1, was examined using IHC. The data indicated that the whole population of macrophages, including activated and resting phases, was reduced after different treatments. Additionally, the expressions of pro-inflammatory genes were downregulated, with the anti-inflammatory genes upregulated in the early phase, as demonstrated in our study. Thirdly, MSCs contribute to angio-genesis. They secrete growth factors, including VEGF, PDGF, IGF, HGF, β-FGF, SDF-1, TGF-β, and GDF11, stimulating neovascularization [19]. Additionally, they stimulate the proliferation of local stem cells, as the number of Sox 9-positive cells increased significantly in our study and promoted neovascularization through the MMPs family. Additionally, the expressions of MMP1, MMP2, and TIMP1, which facilitate the pro-angiogenesis, were altered. Fourthly, MSCs are involved in re-epithelialization and ECM remodeling. In the last phase, collagen fibers re-construct in a proper manner and ECM remodeling occurs. The cytokines released by MSCs are related to different fibroblast characteristics, including cell proliferation, migration, collagen synthesis, and other ECM protein expressions [20]. In our study, more collagen fibers, including collagen I and collagen III, were examined after different treatments. 

### 3.2. The Interaction between MSCs and NF3

Their limited survival time and survival rate restrict the use of MSCs. Different strategies have been investigated to enhance their therapeutic effects. The following methods are commonly used in the published studies: hypoxic pre-conditioning, differentiation prior to engraftment, genetic modification, and priming with other drugs or small molecules [21]. In our study, we used MSCs together with a Chinese herb, aiming to alter their therapeutic benefits regarding promoting wound healing. Meanwhile, our data showed that NF3 prolonged the survival time and survival rate of MSCs after injection, which enhanced their paracrine activity indirectly. A group of proteins were detected in the supernatants of MSCs treated with NF3. According to their function related to wound healing, the proteins could be categorized into inflammation, angiogenesis, and ECM remodeling. After PCR validation, the expressions of MMP3, IGFBP-3, cystatin-C, and LIF increased remarkably. MMP3 is an important factor in the ECM remodeling phase of wound healing. It regulates the rate of wound healing through its role in wound contraction [22]. IGFBP3 is a secreted glycoprotein which presents multiple roles both inside and outside cells. It regulates cell growth and interacts with other cells through different binding proteins [23]. Cystatin-C is a non-glycosylated protein which can be found in virtually all nucleated cells. It is used as a biomarker for renal disease clinically [24]. The upregulation of IGFBP3 and cystatin-c in our study may relate to the further activation of the paracrine activity of MSCs. LIF is a member of the interleukin-6 (IL-6) cytokine family. It has an important role in the development through the JAK-STAT pathway [25]. The increase in LIF after NF3 treatment may indicate an enhanced stemness in MSCs. 

Wnt signaling plays an important role in the regulation of self-renewal and differentiation of MSCs. Most MSCs undergo senescence rapidly after administration, which limits the number of cells promoting tissue regeneration [26]. Wnt signals support the cell expansion and keep cells in an un-differentiated state. In our western blot data, the expression of the Wnt 2 protein in MSCs increased significantly after NF3 treatment, which corresponded to the prolonged survival time and increased survival rate in animal studies, indicating that the Wnt signaling could be a potential target in their interaction. There is crosstalk between Wnt and other growth factor signaling pathways. Together with TGFβ-1, they cooperate to control the self-renewal of MSCs. TGFβ-1 is critical in inducing wound healing through activating MSCs differentiation, increasing their ECM related secretion, and promoting keratinocyte differentiation and maturation [19]. It induced intracellular signaling via Smad 2/3 transcription factors and promoted the expression of collagen I and collagen III, which accelerate re-epithelialization in wound healing [27]. In our study, the expressions of TGFβ-1 and Smad2/3 increased remarkably after NF3 treatment, indicating that TGFβ-1 signaling is another potential target in their interaction. 

### 3.3. Effects of NF3 on FR and RAW 264.7

Fibroblasts are the major cell type responsible for granulation during wound healing. They secrete various growth factors which facilitate angiogenesis, cell proliferation, and ECM remodeling. An appropriate inflammatory response is necessary for wound healing. Macrophage plays an essential role in mediating the process [9,12]. FR is the dermal fibroblast isolated from rat skin. Raw 264.7 is a macrophage cell line isolated from mice. The promotion of cell viability, proliferation, and mobility in FR, and the inhibitory effect in LPS-activated macrophage after NF3 treatment are consistent with our previous study [9,12]. 

### 3.4. Clinical Use of MSCs and NF3

There are more than 50 clinical trials using stem cells for treating different wounds registered on clinicaltrials.gov currently. The wounds include burns, diabetic ulcers, pressure sores, surgical wounds, and hypertrophic scars. Our team has published some studies using MSCs to treat refractory ulcers [28,29]. Additionally, we have recruited some patients with diabetic feet to test the healing effect of NF3 clinically [13]. All clinical data has indicated a promising result. However, there have been no clinical studies on the joint treatment so far.

### 3.5. Limitations to the Study

There are several limitations to this study. Firstly, we applied MSCs and NF3 according to our previous experiences, and different doses were not tested. Secondly, the differentiation of MSCs was not examined after NF3 treatment. Thirdly, the polarization of M1 and M2 macrophages was not investigated. Fourthly, only 79 proteins in the NF3-treated supernatant were examined; more wound healing related proteins, such as EGF, FGF, PDGF, and HGH have not been tested.

In this study, we demonstrated the combined use of MSCs and NF3 on a wound healing model for the first time. The joint treatment showed an enhanced therapeutic effect in the form of anti-inflammation, pro-angiogenesis, and modulation in ECM deposition. This could be a new direction for improving the functions of MSCs in the future.

## 4. Materials and Method

### 4.1. In Vivo Study

#### 4.1.1. Wound Healing Model

All surgical activities were performed regarding the Animals (Control of Experiments) Ordinance Chapter 340, Department of Health, Hong Kong. The study was approved by the Animal Experimentation Ethics Committee of the Chinese University of Hong Kong.

Adult female SD rats weighing 300–350 g were used for the experiment. They were anesthetized using the cocktail of ketamine (50 mg/kg) and xylazine (10 mg/kg) via intra-peritoneal injection. During the surgery, their body temperature was maintained at 37 °C using a warm pad. A 2.5 cm × 2.5 cm full-thickness skin excision was taken from the dorsal area of the rat using scissors and forceps. Four million MSCs (n = 10) were injected at five sites (four surrounding the wound and one in the middle) using an insulin syringe. NF3 was given orally (1 g/kg) (n = 10). The combined treatment consisted of the MSC injections and oral NF3 administration (n = 10). No treatment was given to the control animals (n = 10). After surgery, the wound was covered by sterile Vaseline gauze and all animals were kept in cages individually. After surgery, the wound size was measured every three to four days and recorded as length (cm) × width (cm). 

#### 4.1.2. Immunohistochemistry (IHC) Staining

Animals were sacrificed on Day 7 and Day 14 after surgery. All samples were paraffin embedded and were incised into 5 μm-thick pieces. Five photos were taken under different magnifications (collagen I and collagen III: 100×; CD 31, Sox 9, and ED-1: 400×) at the wound area for examination. 

Mouse monoclonal antibody against collagen I (1:150, Invitrogen, Waltham, MA, USA), collagen III (1:500, Invitrogen, Waltham, MA, USA), Sox-9 (1:50, Abcam, Cambridge, UK), CD31 (1:2000, Abcam, Cambridge, UK), and ED-1 (1:500, Abcam, Cambridge, UK) were used in IHC staining to detect ECM remodeling, local stem cell activation, angiogenesis, and inflammation. After being dewaxed in xylene baths, the slides were stained with primary antibodies, as described above. The secondary antibody was used as a negative control. All slides were treated with diaminobenzidine (DAB) (K346811, DAKO, Glostrup, Denmark) solution for 30 s and the nucleus was stained with hematoxylin. 

#### 4.1.3. Real-Time PCR

Total RNA was isolated using TRIzol* Reagent (Invitrogen, Life Technology, Hong Kong, China). A total of 2 ng of the extracted RNA was used for cDNA synthesis. The PCR cycle was performed under different conditions: at 95 °C for 10 min followed by 40 cycles of 95 °C for 15 s, and 60 °C for 60 s in QuantStudioTM 12K Flex system. The threshold cycle (Ct) of each target gene was defined by the system automatically. Glyceraldehyde 3-phosphate dehydrogenase (GAPDH) was used to normalize the Ct value of the target gene (^Δ^Ct value). For each mRNA, the relative differences in the expression levels were calculated (^ΔΔ^Ct) and presented in fold change (2^−ΔΔ^Ct). The primers are listed in Table 2.

### 4.2. In Vitro Study

#### 4.2.1. Cell Line Culture, and MSC Cultivation and Identification 

Normal rat dermal fibroblast (FR) and murine monocyte-macrophage (RAW 264.7) were purchased from the American Type Culture Collection (ATCC; Manassas, VA, USA). FR and RAW264.7 cells were maintained in high-glucose DMEM (d-glucose: 4500 mg/L; ATCC, Manassas, VA, USA), 10% fetal bovine serum (FBS; GIBCO, New York, USA), and 1% penicillin–streptomycin (PS; GIBCO, New York, NY, USA). The cells were subcultured until they reached 80% confluency. 

Adult male transgenic Sprague-Dawley (SD) rats (300–350 g) that expressed green fluorescent protein (GFP) (SDTg(CAG-EGFP) CZ-0040sb; SLC Inc, Shizuoka, Japan) were used as donors to harvest the adipose tissue from, as previously described [9,14,17]. The animals underwent anesthesia via intraperitoneal injection (i.p.) of a ketamine (50 mg/kg) and xylazine (10 mg/kg) mixture. The subcutaneous adipose tissue at the inguinal region was exposed and collected for cultivation. After being washed with sterile PBS, the tissue was treated with 0.1% collagenase (type I; Sigma-Aldrich, Shanghai, China) in Dulbecco’s modified Eagle’s medium (DMEM, Invitrogen, Life Technology, Hong Kong) supplemented with 100 units/mL penicillin, 100 μg/mL streptomycin, and 2 mM L-glutamine (Invitrogen, Life Technology, Hong Kong, China) for 40 min at 37 °C with gentle agitation. After processing, the cells were seeded in a 75 cm^2^ culture flask (Thermo Fisher Scientific, Hong Kong, China) for further experiment.

After cultivation, ADMSCs were collected and characterized via flow cytometry with a FACS argon laser (BD Bioscience, San Jose, CA, USA). They were labelled with phycoerythrin-conjugated antibodies against CD29, CD45, and CD90 (Abcam Inc., Cambridge, UK) [9]. Isotype-matched negative controls were used to evaluate the background fluorescence.

To identify the differentiation potential, ADMSCs were cultured in adipogenic, chondrogenic, and osteogenic differentiation culture medias according to the manufacturer’s protocols (Invitrogen, LifeTechnology, Hong Kong, China). The differentiated adipocytes were stained with Oil Red O, the chondrocytes with Alcian Blue, and the osteocytes with Alizarin Red S stain to identify intracytoplasmic lipid, extracellular glycosaminoglycans, and calcium deposits, respectively. All chemicals were purchased from Sigma-Aldrich, Shanghai, China [15].

#### 4.2.2. NF3 Preparation, Administration, and Supernatant Collection

NF3 consisted of two herbs Astragali Radix (AR) and Rehmanniae Radix (RR) at a ratio of 2:1. The raw herbs were purchased from mainland China and authenticated via morphological characterizations and thin layer chromatography in accordance with the Chinese Pharmacopoeia. Voucher specimens of RA and RR were deposited in the museum of Institute of Chinese Medicine, at the Chinese University of Hong Kong, with the voucher specimen numbers 2008–3201 for AR and 2008–3200 for RR [12]. The two herbs were cut into small pieces and mixed for extraction. After being soaked in 10 volumes of distilled water for 30 min, the herbs were boiled under reflux for 1 h twice. The extracts were pooled, filtered, and lyophilized into dry powder. The extraction yield was around 34% (*w*/*w*) [12]. Different doses of NF3 (0.025–2 mg/mL) were added to the culture flasks of MSCs, FR (dermal fibroblasts), and RAW 264.7 (macrophages) for 24 h. After treatment, the supernatants of MSCs and RAW 264.7 were collected for further experiment. The effect of NF3 on the three cell types were studied as described below.

The total protein concentration in the supernatant was analyzed by Bicinchoninic acid (BCA) assay. The culture media was changed to serum-free DMEM before NF3 application to exclude the impact of FBS. After 24 h, the serum-free supernatant was collected. Bovine serum albumin (BSA) was used as standard. The optical absorbance was measured at 562 nm with a microplate reader. The protein concentration was calibrated through the plot of standard BSA concentrations. Each sample was tested in triplicate. 

#### 4.2.3. Proteome Profiler Array

A total of 79 soluble proteins including cytokines and chemokine can be detected by the Proteome Profiler Rat XL Cytokine Array (R&D Systems, Inc., Hong Kong, China) (Appendix A). Dulbecco’s modified Eagle’s medium (DMEM, Invitrogen, Life Technology, Hong Kong, China) was used as a negative control. After blocking the membranes, 1× Streptavidin-HRP was added as a secondary antibody. A total of 1 mL of Chemi Reagent Mix was added to each membrane after incubation at room temperature for 30 min. To get the images, the membranes were placed in an autoradiography film cassette and exposed to X-ray for 5 min. The proteins with positive signals on the developed film were identified by placing the transparency overlay template on the image. ImageJ 1.52v software was used to compare the density of positive signals on a different membrane.

#### 4.2.4. Western Blot

Proteins extracted from 1 × 10^6^ MSCs treated with different doses of NF3 (0.25–2 mg/mL) were used for the western blot. Cells were washed with PBS and lysed with a whole-cell extraction buffer (2% SDS, 10% glycerol, 625 mM Tris–HCl, pH 6.8) for 30 min at 4 °C. Then they were centrifuged at 14,000× *g* for 15 min at 4 °C and collected. Samples were heated at 95 °C for 5 min, resolved by 10% SDS polyacrylamide gel, and transferred to 0.45-mm PVDF membrane (Immobilon, Millipore, Burlington, MA, USA) at 120 V for 1.5 h. The membranes were washed with blocking buffer of 5% non-fat dried milk in Trisbuffered saline Tween 20 (TBST). The membranes were incubated with primary antibodies: beta-actin (Sigma, Irvine, CA, USA), wnt2, β-catenin, TGFβ-1, and Smad2/3 (Abcam, Cambridge, UK), and subsequently incubated with secondary horseradish peroxidase-conjugated antibodies (Invitrogen, Waltham, MA, USA). Following the addition of ECL solution (GE Healthcare Life Sciences, Uppsala, Sweden), the images were captured using ChemiDoc XRS+ (Bio-Rad, Hercules, CA, USA), and analyzed by Image J 1.52v software.

#### 4.2.5. Cell Viability: MTT

The cell viability of MSCs, FR, and RAW 264.7 were tested using a 3-[4,5-dimethylthiazol-2-yl]-2,5-diphenyl-tetrazolium bromide (MTT) (Sigma, Irvine, CA, USA) assay. The cells were seeded in 96-well plates (MSCs: 5 × 10^3^/well; FR: 5 × 10^3^/well; RAW 264.7: 1 × 10^5^/well) for 24 h. Different doses of NF3 (0.025–2 mg/mL) were added and the mixtures were incubated for another 24 h. On the third day, the cell viability was measured by recording a reduction in MTT dye in living cells to blue formazan crystals at an optical density of 540 nm. Cells cultured in Dulbecco modified Eagle medium (DMEM) were used as control.

#### 4.2.6. Cell Proliferation: BrdU

A BrdU proliferation assay was used to detect MSCs and FR proliferation after being treated with different doses of NF3 (0.025–2 mg/mL) for 24 h. All procedures were performed according to the manufacturer’s instructions (Sigma, Irvine, CA, USA). Cells were incubated in BrdU-labelling solution for 4 h. After being fixed and denatured by FixDenat, cells were incubated with anti-BrdU antibodies for 90 min. Substrate solution was used for color reaction and the optical density was measured at 450 nm. Cells cultured in Dulbecco modified Eagle medium (DMEM) were used as control. Medium without cells was used as blank control.

#### 4.2.7. Cell Mobility: Scratch Test

The migration of MSCs and FR was examined using scratch test. A total of 5 × 10^4^ of MSCs/FR was seeded in each well of a 24-well plate supplied with low glucose DMEM (Gibco, New York, NY, USA), 10% FBS (Gibco, New York, NY, USA), and 1% penicillin–streptomycin (PS) (Invitrogen Co., Carlsbad, CA, USA). After the cell reached high confluency, the medium was replaced with DMEM containing 1% FBS, and the cells were starved for 24 h. Then two crosses were scrapped into each well using 200 μL p tips. After photos of the two crosses were taken under microscope at 4× magnification, different doses of NF3 (0.025–2 mg/mL) were added to each well. Normal DMEM was used as control. The images were analyzed using TScratch 1.0 software. The percentage of closed area was compared with the area before treatment and recorded as relative wound area.

#### 4.2.8. Nitric Oxide (NO) Inhibitory Assay

Lipopolysaccharides (LPS) was used to induce inflammatory response in RAW264.7 (macrophages), and the severity was shown by the concentration of nitric oxide (NO). RAW 264.7 were seeded at the density of 4 × 10^5^ per well in a 24-well plate for overnight incubation. Different doses of NF3 (0.025–2 mg/mL/) with 1 μg/mL LPS were added to each well and incubated for 24 h. After incubation, 100 μL of cell supernatant was transferred to a 96-well plate. A total of 100 μL Griess Reagent was added to each well. After 10 min, the plate was read at 540 nm in a microplate reader. The standard curve was plotted by the given concentration of sodium nitric (NaNO3). DMEM was used as negative control.

#### 4.2.9. Enzyme-Linked Immunoassay (ELISA) 

The concentrations of pro-inflammatory cytokines IL-6, MCP-1, and TNF-α in RAW 264.7 supernatant was measured using ELISA kits (BD OptEIA™, BD Biosciences, San Diego, CA, USA). All procedures were performed according to the manufacturer’s instructions.

### 4.3. Statistical Analysis

All data are expressed in mean ± standard error of the mean (SEM). All data underwent one-way or two-way analysis of variance (ANOVA) using GraphPad Prism 5. A *p* value of less than 0.05 was considered significant. Dunnett test was applied as a post-hoc comparison.

## Figures and Tables

**Figure 1 ijms-24-01372-f001:**
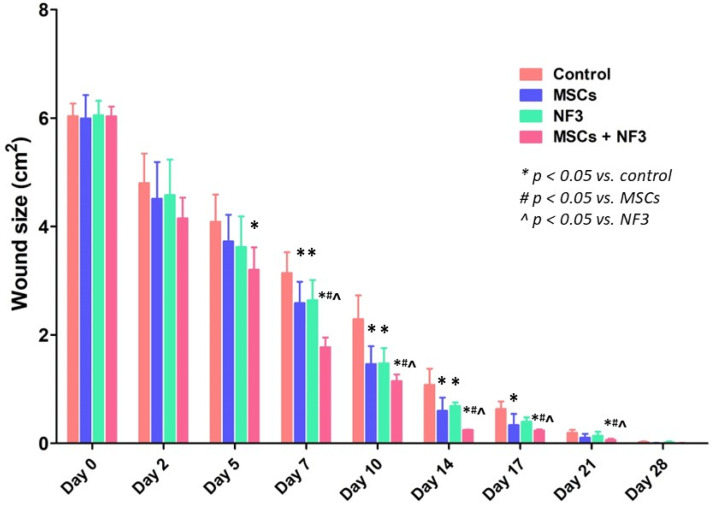
Wound size at different time points in each group. In the MSCs + NF3 group, the wound size presented a significant reduction from Day 5 onwards (*p* < 0.05). On Day 7, 10, and 14, all treatments promoted wound healing significantly (*p* < 0.05), while the combined treatment further enhanced the healing effect (*p* < 0.05). On Day 17, MSCs and the combined treatment kept reducing the wound size (*p* < 0.05), but no significant difference was observed in the NF3 group. The wound was closed and completely healed by Day 21 in the MSCs + NF3 group (*p* < 0.05). All wounds had completely healed by Day 28.

**Figure 2 ijms-24-01372-f002:**
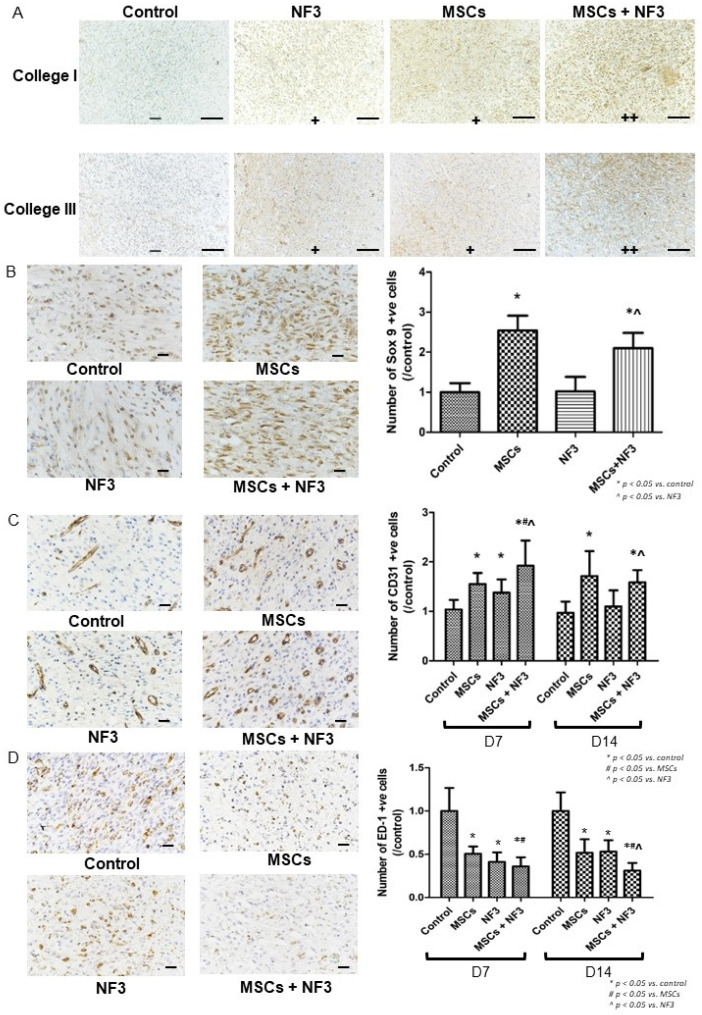
The representative images of IHC staining and their corresponding quantitative results. (**A**) Representative images of anti-collagen I and anti-collagen III staining (magnification: 100×, scale bar: 100 µm, − negative, + positive, ++ strong positive). The upper image is anti-collagen I and the lower is anti-collagen III. From left to right, the expression of the two collagens increased, and the most collagen formation was detected in the MSCs + NF3 group. (**B**) Anti-Sox 9 staining (magnification: 400×, scale bar: 100 µm). On the left is a representative photo of Sox 9-positive cells. Sox 9 is a marker of stem cells. We only stained the tissue collected on Day 14 to exclude the injected MSCs. The quantified data on the right indicated both MSCs and the combined treatment activated or recruited more Sox 9-positive cells (*p* < 0.05). No significant difference was observed in the NF3 group. (**C**) Anti-CD31 staining (magnification: 400×, scale bar: 100 µm). CD31 is a marker of neovascularization. On the left are representative images of CD31 positive cells. The graph on the right indicates that both MSCs and the combined treatment increased the number of CD31 positive cells on Day 7 and Day 14, with the highest number of positive cells detected in the combined group (*p* < 0.05). In the NF3 group, this increase was only observed on Day 7 (*p* < 0.05). (**D**) Anti-ED1 staining (magnification: 400×, scale bar: 100 µm). ED-1 is a pan-marker of macrophage, which stains both activated and resting types. On the left are representative images of ED-1 positive cells. The graph on the right shows that all treatments reduced the number of ED-1 positive cells significantly on both Day 7 and Day 14, while the combined treatment further enhanced this effect (*p* < 0.05).

**Figure 3 ijms-24-01372-f003:**
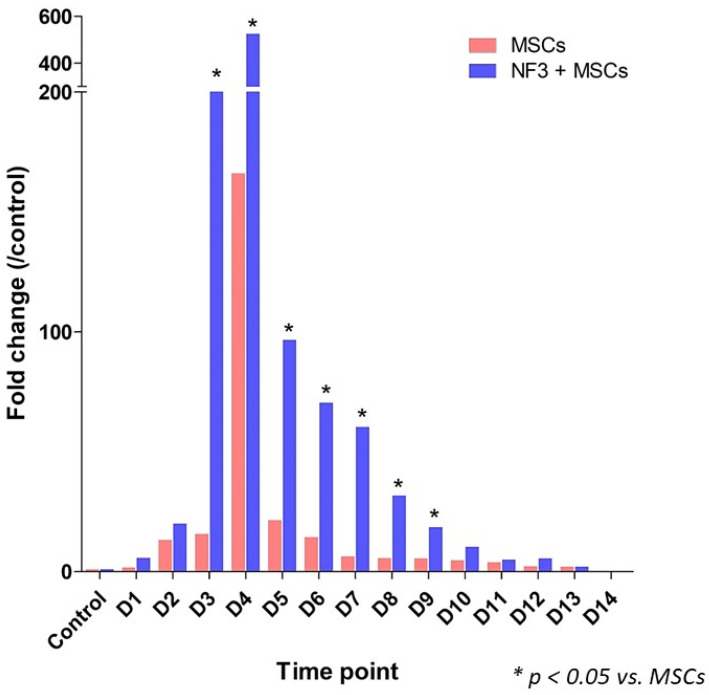
The expression of GFP in skin examined by RT-PCR. After GFP-MSC injection, the skin was harvested daily for analysis. The peak level of expression presented on Day 4 in both groups. It dropped gradually from Day 5 and diminished on Day 14. Compared with the MSCs group, a significant increase in the GFP expression was observed in the combined group from Day 3 to Day 9. No GFP expression was detected on Day 14.

**Figure 4 ijms-24-01372-f004:**
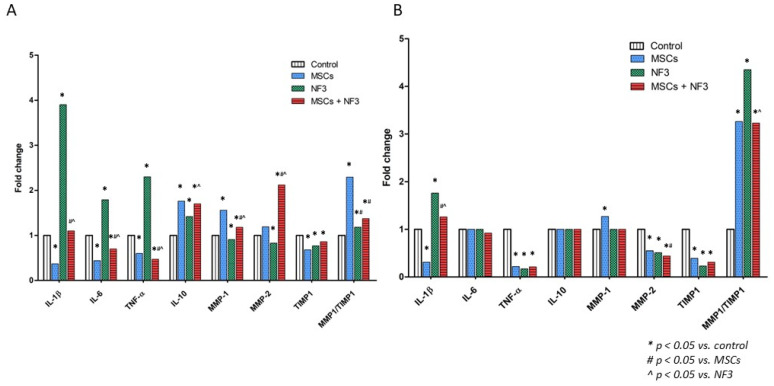
The expression of inflammation-related and ECM-related genes in skin. The tissues were collected 7 days and 14 days after surgery. (**A**) Gene expressions on Day 7. The expressions of anti-inflammatory genes (IL-1β, IL-6, and TNF-α) were downregulated in both the MSCs and MSCs + NF3 groups (*p* < 0.05). Their expressions were upregulated in the NF3 group. The expression of the anti-inflammatory gene (IL-10) was upregulated significantly in all groups (*p* < 0.05). In terms of ECM-related genes, the expressions of MMP-1 and MMP-2 were upregulated significantly in the MSCs and MSCs + NF3 groups (*p* < 0.05), while it was downregulated in the NF3 group. The expression of its inhibitor, TIMP1, was downregulated significantly in all groups (*p* < 0.05). After calculating the ratio of MMP1 to TIMP1, it was clear that this increased significantly in all groups (*p* < 0.05). (**B**) Gene expression on Day 14. The expression of IL-1β was downregulated significantly in the MSCs group (*p* < 0.05). Its expression was upregulated in the NF3 and MSCs + NF3 groups (*p* < 0.05). The expression of TNF-α was downregulated remarkably in all groups (*p* < 0.05). No difference was observed in the expression of IL-6 or IL-10. The expression of MMP-1 was upregulated only in the MSCs group (*p* < 0.05). The expression of MMP-2 and TIMP-1 was downregulated significantly in all groups (*p* < 0.05). After calculating the ratio of MMP1 to TIMP1, it was clear that this increased significantly (*p* < 0.05).

**Figure 5 ijms-24-01372-f005:**
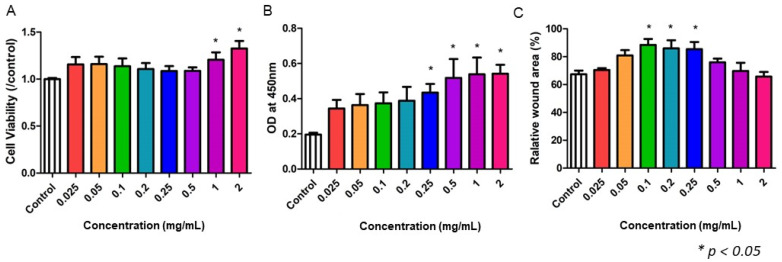
The effect of NF3 on MSCs. Different doses of NF3 were supplied to MSCs cultivation. (**A**) MTT. The data showed that a high dose of NF3 (1–2 mg/mL) promoted cell viability in MSCs (*p* < 0.05). (**B**) BrdU. NF3 (0.5–2 mg/mL) enhanced the proliferation of MSCs. (**C**) Scratch test. A medium dose of NF3 (0.1–0.25 mg/mL) increased the mobility of MSCs.

**Figure 6 ijms-24-01372-f006:**
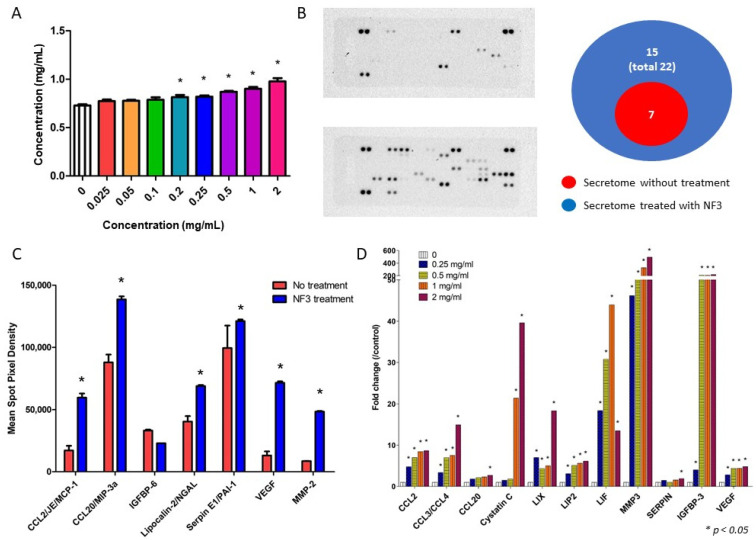
Results of protein array. A total of 79 proteins were examined in the supernatants of MSCs with or without NF3 treatment. (**A**) Total protein concentration. This indicated that after different doses of NF3 (0.2–2 mg/mL) treatment, the protein concentration in the supernatant was increased. (**B**) Results of Proteome Profiler Rat XL Cytokine Array. Left: an image of the membrane in the array. The upper image shows the untreated supernatant and the lower shows the NF3-treated supernatant. As shown in the diagram on the right, there were seven proteins detected in the untreated supernatant, while twenty-two were detected in the NF3-treated supernatant. (**C**) Semi-quantitative analysis of the protein expression. The data showed that the expressions of the most common proteins (except for IGFBP-6) were upregulated after NF3 treatment. (**D**) RT-PCR validation. The expressions of 22 proteins were validated by RT-PCR. The expressions were upregulated significantly after different doses of NF3 treatment (0.25–2 mg/mL) and presented in a dose-dependent manner.

**Figure 7 ijms-24-01372-f007:**
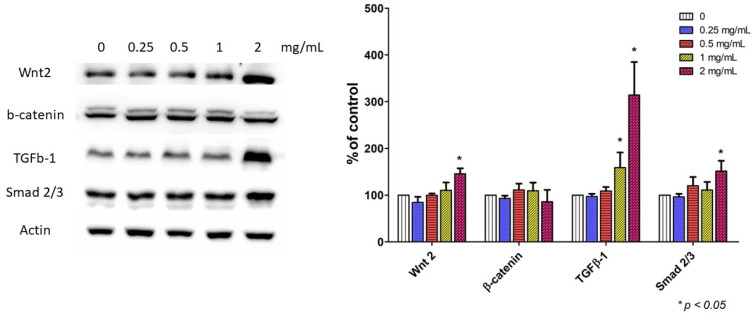
The results of western blot. (**Left**): an image of the western blot. As the dose of NF3 increased (0.25–2 mg/mL), the expressions of Wnt 2, TGFβ-1, and Smad 2/3 increased. (**Right**): the quantitative result. Compared with the control, the expressions of Wnt 2, TGFβ-1, and Smad 2/3 increased significantly after NF3 treatment (*p* < 0.05).

**Figure 8 ijms-24-01372-f008:**
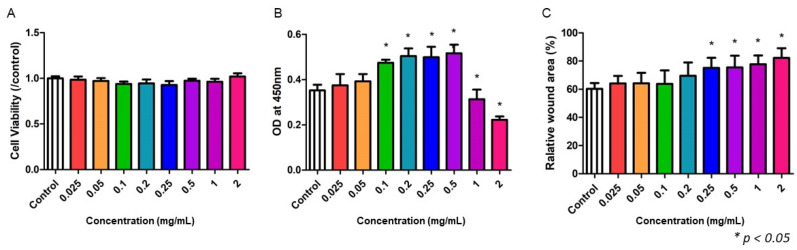
The effect of NF3 on FR. Different doses of NF3 (0.25–2 mg/mL) were added to FR cultivation. (**A**) MTT. No cytotoxic effect was observed in any of the doses. (**B**) BrdU. The proliferation ability was enhanced when a medium dose of NF3 was supplied (0.1–0.5 mg/mL) (*p* < 0.05). As the dose increased (1–2 mg/mL), the proliferation was suppressed significantly (*p* < 0.05). (**C**) Scratch test. A high dose of NF3 (0.25–2 mg/mL) promoted the cell mobility in FR (*p* < 0.05).

**Figure 9 ijms-24-01372-f009:**
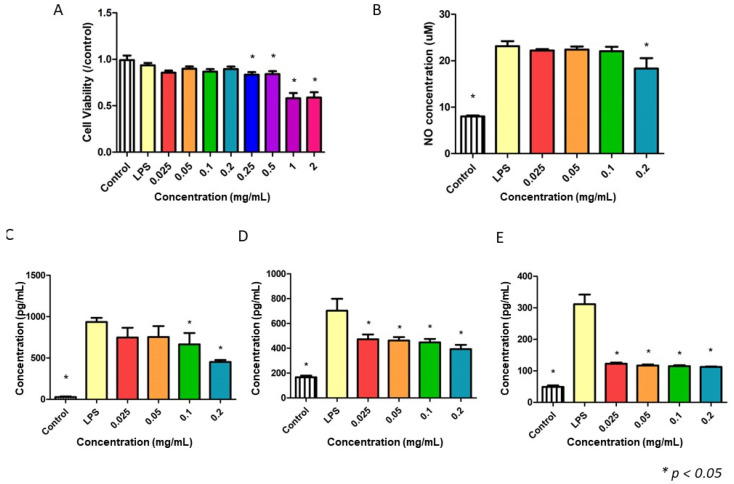
The effects of NF3 on RAW 264.7. Different doses of NF3 (0.25–2 mg/mL) were added to macrophage cultivation. RAW 264.7 was activated by 1 µg/mL LPS. (**A**) MTT. The data showed that a high dose of NF3 (0.25–2 mg/mL) reduced cell viability in RAW 264.7 (*p* < 0.05). (**B**) Nitric oxide (NO) inhibitory assay. The level of NO was suppressed when 0.2 mg/mL NF3 was supplied (*p* < 0.05). (**C**–**E**) ELISA results of IL-6, MCP-1, and TNF-α, respectively. NF3 (0.025–0.2 mg/mL) reduced the levels of pro-inflammatory cytokines in the supernatant of LPS-activated macrophages (*p* < 0.05).

**Table 1 ijms-24-01372-t001:** List of proteins detected in the supernatants of MSCs. The upper section details the seven common proteins that existed in both NF3-treated and un-treated supernatant. The lower section details the 15 unique proteins detected only in NF3-treated supernatant. Their functions are related to inflammation, angiogenesis, ECM remodeling, apoptosis, and cell signaling. In terms of cell signaling, this includes cell adhesion, signal transduction, migration, proliferation, differentiation, senescence, etc.

	Inflammation	Angiogenesis	ECM Remodelling	Apoptosis	Cell Signalling
Function of 7 Common Proteins					
CCL2	✓				
CCL20	✓				
IGFBP-6					✓
MMP-2	✓	✓			
Serpin El			✓		
Lipocalin-2	✓				
VEGF		✓	✓		
Function of Other Proteins					
CCL3	✓				
OCC12	✓				
CCLI1	✓				
Cystatin C					✓
IGFBP-3					✓
Galectin-3				✓	✓
G-CSF				✓	
IL-6	✓				
MMP-3		✓	✓		
WISP-1					✓
NOV				✓	✓
Osteopontin				✓	✓
Osteoprotegrin				✓	✓
LIF					✓
LIX	✓				

**Table 2 ijms-24-01372-t002:** The primers used in RT-PCR.

Gene	Forward	Backward
IL-1β	CAC CTT CTT TTC CTT CAT CTT TG	GTC GTT GCT TGT CTC TCC TTG TA
IL-6	TGA TGG ATG CTT CCA AAC TG	GAG CAT TGG AAG TTG GGG TA
TNF-α	ACT GAA CTT CGG GGT GAT TG	GCT TGG TGG TTT GCT ACG AC
IL-10	TGCCTTCAGTCAAGTGAAGAC	AAACTCATTCATGGCCTTGTA
MMP1	CCACTAACATTCGAAAGGGTTT	GGTCCATCAAATGGGTTATTG
MMP2	AAAGGAGGGCTGCATTGTGAA	CTGGGGAAGGACGTGAAGAGG
TIMP1	CAGCAAAGGCCTTCGTAAA	TGGCTGAACAGGGAAACACT
CCL2	CTG TCT CAG CCA GAT GCA GTT	GAG CTT GGT GAC AAA TAC TAC A
CCL3	CATGGCGCTCTGGAACGAA	TGCCGTCCATAGGAGAAGCA
CCL4	CCAATAGGCTCTGACCCTCC	AAAGGCTGCTGGTCTCATAGT
CXCL2	TCC TCA ATG CTG TAC TGG TCC	ATG TTC TTC CTT TCC AGG TC
CCL11	GCCATAGTCTTCAAGACCAAGCTT	TGGCATCCTGGACCCACTT
Cystatin C	AGG AGA AGA GAA CCA GGG GAC AGC	AGT ACA ACA AGG GCA GCA ACG ATG
CCL20	GACTGCTGCCTCACGTACAC	CGACTTCAGGTGAAAGATGATAG
IGFBP-3	ACAGCCAGCGCTACAAAGTT	GCGGTATCTACTGGCTCTGC
Galectin-3	CGGGATCCAGGAAAATGGCAGACGGCTTC	GGGGTACCTCATAACACACAGGGCAGTTC
G-CSF	TTGCCACCACCATCTGGC	ACTGCTGTTTAAATATTAAACAGGG
IGFBP-6	CCGTCGGAAGAGACTACCAAG	CTTGAACAGGACTGGGCCTT
MMP-3	CAGGCATTGGCACAAAGGTG	GTGGGTCACTTTCCCTGCAT
WISP-1/CCN4	AGAGCCGCCTCTGCAACTT	GGAGAAGCCAAGCCCATCA
NOV/CCN3	CTACAGAGTGGAGCGCGTGTT	GGAAGATTCCTGTTGGTGACCC
Serpin E1/PAI-1	TCTGGGAAAGGGTTCACTTTACC	GACACGCCATAGGGAGAGAAG
Osteopontin (OPN)	CCAGCACACAAGCAGACGTT	TCAGTCCATAAGCCAAGCTAT
Osteoprotegrin/TNFRSF11B	TGGCACACGAGTGATGAATGCG	GCTGGAAAGTTTGCTCTTGCG
LIF	CATGACGGATTTCCCACCTTT	GCAGCCCAACTTCTTCCTTTG
Lipocalin-2/NGAL	GGAATATTCACAGCTACCCTC	TTGTTATCCTTGAGGCCCAG
VEGF	ACAGAAGGGGAGCAGAAAGCCCAT	CGCTCTGACCAAGGCTCACAGT
LIX	CTCAAGCTGCTCCTTTCTCG	GCGATCATTTTGGGGTTAAT

## Data Availability

All the relevant data presented in this study are available in the manuscript. The raw data are available on request from the corresponding author.

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
