# Peer review of "The Application of Adipose Tissue-Derived Mesenchymal Stem Cells (ADMSCs) and a Twin-Herb Formula to the Rodent Wound Healing Model: Use Alone or Together?"

_ijms, 2023, doi:10.3390/ijms24021372_

Round 1

Reviewer 1 Report

This study has reported mesenchymal stem cells (MSCs) accelerated the wound healing process  in a rodent skin excision model. The research focusing on the interaction of MSCs and Chinese medicine is limited. The authors applied MSCs and the twin herb formula in the wound healing model and investigated their interactions. And the results suggested NF3, a twin-herb formula, enhanced the therapeutic effect of MSCs in wound healing model and the TGFβ/Smad and Wnt pathways were related to the procedure for the first time.

However, there are key issues that the authors must address before the manuscript can be considered for publication:

 1. Please add scale bar for figure 2A,B,C.

 2. All histogram formats are not uniform, some are color, some are black and white, please unify.

Reviewer 2 Report

The paper of Ma et al. presents data on the effectiveness in a rodent wound healing assay of the treatment with MSC, an herbal extract (NF3), and a combination of both. The treatments improved wound healing in all groups, but the combination had more effects on the expression of collagens, SOX9, and on the downregulation of proinflammatory cytokines, and on upregulation of antinflammatory cytokines. The paper is interesting, but I have few questions to the authors:

11) Paragraph 2.3: the authors talk about the expression of GFP protein by RT-PCR, but in materials the is no mention of the use of any kind of cell expressing GFP, so please add what kind of cell are, how are used and any reference

22) Paragraph 2.6: the authors stated that they measured total protein concentration in the supernatant of MSC colture treated with NF3, this mean that they measured also the fetal calf serum added in the colture, please explain in the text and in materials

33)      Paragraph 2.8: the authors mention about the cell line FR (dermal fibroblast), but there is nothing about them in materials

44)      Figure 8: please correct median dose, in statistic median have a different meaning; please correct the X axis in figure 8c

55)      Paragraph 2.9: please add some details about the colture of RAW364.7 cell line

66)      Paragraph 3.2: the authors stated that the upregulation of cystatin and IGFBP3 could be connected to the paracrine acitvity of MSC but this is a speculative statement the authors did not show any proof. How can they explain the increased expression of WNT in WB with the limited survival rate of MSC?

77)      Paragraph 4.2.1: the authors labelled MSC with PE-conjugated antibodies and then what happened? The sentence seems incomplete. Please show the pictures of the differentiation of the MSC.

88)      Please review the English form
